# Complete genome sequencing of SARS-CoV-2 strains that were circulating in Uzbekistan over the course of four pandemic waves

**Gulnoza Esonova**[1]*, **Abrorjon Abdurakhimov**[1,2], **Shakhnoza Ibragimova**[1], **Diyora Kurmaeva**[1], **Jakhongirbek Gulomov**[1], **Doniyor Mirazimov**[3], **Khonsuluv Sohibnazarova**[1], **Alisher Abdullaev**[1], **Shahlo Turdikulova**[1], **Dilbar Dalimova**[1]

**1** Laboratory of Biotechnology, Center for Advanced Technologies under the Ministry of Higher Education, Science and Innovations, Tashkent, Uzbekistan, **2** Laboratory of Genomics, Institute of Biophysics and Biochemistry under National University of Uzbekistan, Tashkent, Uzbekistan, **3** State Institution Zangiota No. 2 special hospital for the treatment of patients with coronavirus infection, Tashkent, Uzbekistan

* esonovagulnoza96@gmail.com

**Data Availability Statement:** The data generated and analysed in this study were submitted to GISAID and available for registered users at https://

## Abstract

Since the rapid emergence of severe acute respiratory syndrome coronavirus 2 (SARS-CoV-2) as a global COVID-19 pandemic affecting millions of people globally, it has become one of the most urgent research topics worldwide to better understand the pathogenesis of COVID-19 and the impact of the harmful variants. In the present study, we conducted whole genome sequencing (WGS) analysis of 110 SARS-CoV-2 genomes, to give more data about the circulation of SARS-CoV-2 variants during the four waves of pandemic in Uzbekistan. The whole genome sequencing of SARS-CoV-2 samples isolated from PCR-positive patients from Tashkent, Uzbekistan, in the period of 2021 and 2022 were generated using next-generation sequencing approaches and subjected to further genomic analysis. According to our previous studies and the current genome-wide annotations of clinical samples, we have identified four waves of SARS-CoV-2 in Uzbekistan between 2020 and 2022. The dominant variants observed in each wave were Wuhan, Alpha, Delta, and Omicron, respectively. A total of 347 amino acid level variants were identified and of these changes, the most frequent mutations were identified in the ORF1ab region (n = 159), followed by the S gene (n = 115). There were several mutations in all parts of the SAR-CoV-2 genomes but S: D614G, E: T9I, M: A63T, N: G204 R and R203K, NSP12: P323L, and ORF3a(NS3): T223I were the most frequent mutations in these studied viruses. In our previous study, no mutation was found in the envelope (E) protein. In contrast, in our present study, we identified 3 (T9I, T11A and V58F) mutations that made changes to the structure and function of the E protein of SARS-CoV-2. In conclusion, our findings showed that with the emergence of each new variant in our country, the COVID-19 pandemic has also progressed. This may be due to the considerable increase in the number of mutations (Alpha—46, Delta- 146, and Omicron—200 mutations were observed in our samples) in each emerged variant that shows the SARS-CoV-2 evolution.

gisaid.org, with accession IDs from EPI_ISL_18378686 to EPI_ISL_3189001. The whole genome sequencing reads of 110 SARS-CoV-2 samples and genome metadata used and analysed in this study are available in the S1 and S2 Files.

**Funding:** This study has been supported by the research grant from the Ministry of Innovative Development, Republic of Uzbekistan (Research Grant number: -IRV-2021-125). The funders had no role in study design, data collection and analysis, decision to publish, or preparation of the manuscript. There was no additional external funding received for this study and no authors received award from this foundation.

**Competing interests:** The authors have declared that no competing interests exist.

## Introduction

In December 2019, severe acute respiratory syndrome coronavirus 2 (SARS-CoV-2) was first identified in the city of Wuhan in Hubei Province, China which has been known as the cause of the coronavirus disease 2019 (officially declared as the COVID-19 pandemic by the World Health Organization (WHO) on March 11, 2020) [1–4]. Since its first detection, SARS-CoV-2 has spread rapidly to all corners of the world, and by October 31, 2023, 771,549,718 cases of SARS-CoV-2 with 6,974,473 deaths have been reported worldwide (https://covid19.who.int/), while 253,662 confirmed cases with 1,637 deaths (https://www.worldometers.info/coronavirus/country/uzbekistan/) have been reported in Uzbekistan.

The first full-length genome sequences of the novel virus were obtained from five patients at an early stage of the outbreak through metagenomic approaches, supplemented by PCR and Sanger sequencing (January 5, 2020). The genome sequence of Wuhan-Hu-1 (reference genome) was submitted to NCBI/GenBank (GenBank: MN908947) on the same day and released on the open-access virology website (https://virological.org) on January 11, 2020 [1, 2, 5, 6]. As sequencing is essential for epidemiological monitoring, 15,983,874 viral genome sequences of SARS-CoV-2 have been generated and submitted to the online database GISAID (Global Initiative on Sharing All Influenza Data) by 220 countries and territories, as of 31st October 2023 [7].

Over time, SARS-CoV-2 had quickly begun to mutate resulting in sequence diversity and emerging new variants, some of which have been classified by WHO as variants of concern (VOCs): Alpha, first detected in the United Kingdom, in 2020, Beta in South Africa in 2020, Gamma in Brazil in 2020, Delta in India in 2020, and Omicron in South Africa in 2021 [8–10]. Based on the analysis of complete or near-complete viral genomes (sometimes based only on the spike gene of SARS-CoV-2), multiple different nomenclatures have been used to classify SARS-CoV-2. One of the most popular systems for classifying and naming genetically distinct lineages of SARS-CoV-2 used by researchers and public health authorities worldwide is the PANGOLIN (phylogenetic assignment of named global outbreak lineages) nomenclature. The Pango nomenclature uses letters and numbers and is divided into lineages (A, B, B.1, B.1.1, B.1.177, and B.1.1.7.) and sublineages. The following Pango lineages: B.1.1.7, B.1.351, P.1, B.1.617.2, and B.1.1.529 correspond to the Alpha, Beta, Gamma, Delta, and Omicron variants of concern (VOCs), respectively [11]. Based on the GISAID nomenclature, SARS-CoV-2 was also classified into different clades: S, L, V, G, and later G into GH, GR, GV, and more recently GR into GRY and GRA [7]. According to the Nextstrain clade naming strategy SARS-CoV-2 strains were divided into 19A, 19B, 20A, 20B, 20C, 20D, 20E, 20F, 20G, 20H, 20I, 20 J [12, 13].

Since the first confirmed case was reported on March 15, 2020, four waves of the pandemic were recorded in Uzbekistan Fig 1 (https://www.worldometers.info/). The first peak occurred in July—August 2020, when the total number of cases increased rapidly from 12 295 to 43 476, with an increase of 31 181 within two months. As a continuation of the first wave, the second wave occurred in October-November 2020.

During these two peaks, the number of deaths was 299 and 367, respectively, representing 40.7% of the total number of deaths since the beginning of the pandemic in Uzbekistan. The third wave occurred in July and August 2021, with a high mortality rate of 59,3% (n = 971) when the number of infected individuals increased from 113 559 to 160 589 (n = 47 030) within two months, Fig 2 (https://www.worldometers.info). Since the beginning of January 2022, the next-fourth outbreak was observed in the country.

Timely identification of new variants is an important requirement for national health policy, as it enables rapid tracking and investigation of infections in hospitals and communities. This is also particularly important for enhancing vaccination strategies through the timely

## Daily New Cases in Uzbekistan

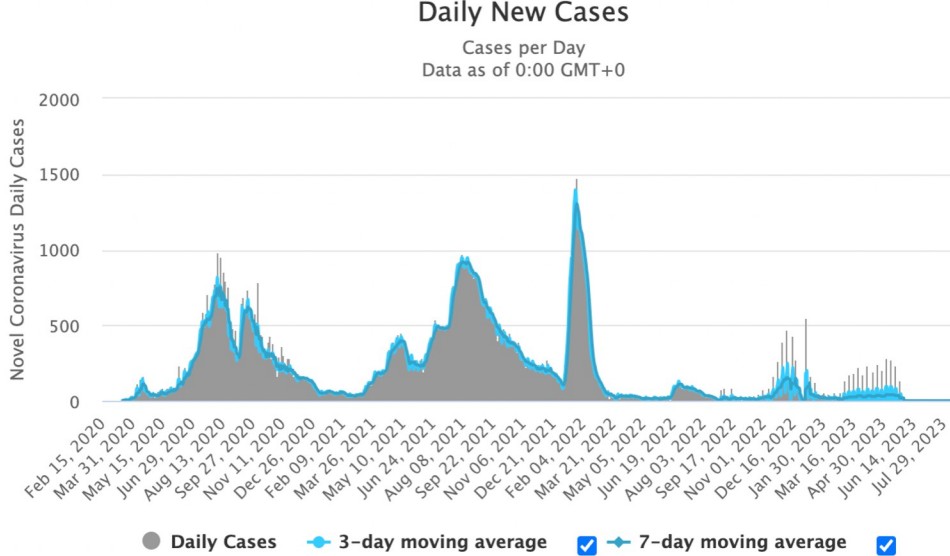

**Fig 1. COVID-19 cases in Uzbekistan since February 2020 to July 2023 (according to Worldometer, https://www. worldometers.info).**

## Daily New Deaths in Uzbekistan

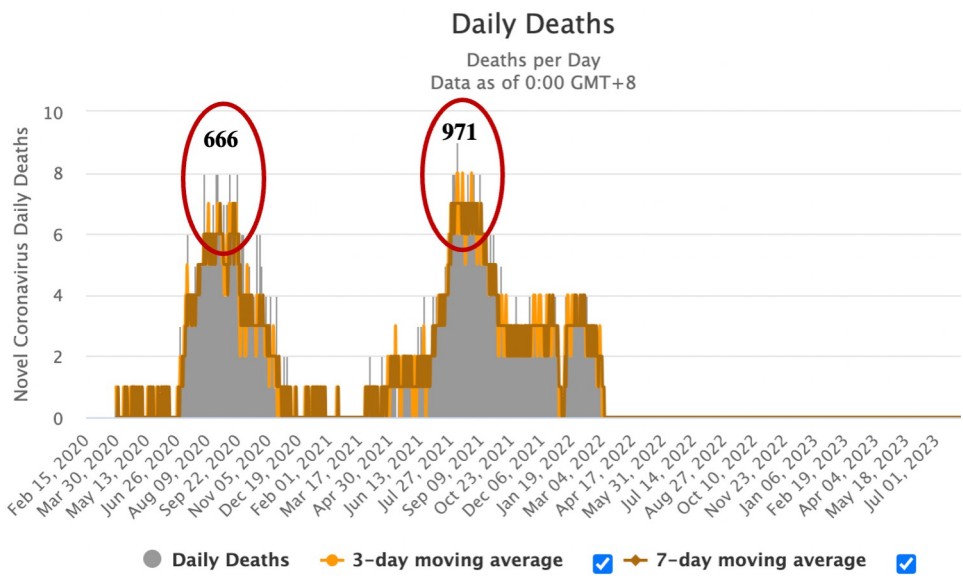

**Fig 2. Mortality from COVID-19 in Uzbekistan from February 2020 to July 2023 (according to Worldometer, Fig 2 (https://www.worldometers.info).**

administration of booster doses to mitigate the effects of waning immunity after vaccination, especially in response to the emergence of new variants [14].

As a continuation of previous studies [15, 16], in this study, we focused on the whole genome sequencing analysis of 110 SARS-CoV-2 genomes to provide insights into the spread of infection, evolutionary patterns, and genetic diversity of the virus to enable effective management and preventive measures in Uzbekistan. Here we report on the four waves of SARS-CoV-2 in the country between 2020 and 2022, with the Wuhan, Alpha, Delta, and Omicron variants dominating in each wave, respectively. We identified a total of 347 amino acid variations in the sequenced genomes, with the most common mutations occurring in the ORF1ab region (159) and the S gene (115). While mutations were present throughout the SARS-CoV-2 genomes, certain mutations such as S: D614G, E: T9I, M: A63T, N: G204R and R203K, NSP12: P323L, and ORF3a(NS3): T223I were most prevalent in the studied genomes. In our previous study, no mutation was found in the envelope (E) protein. In contrast, in our present study we identified 3 (T9I, T11A, and V58F) mutations that made changes to the structure and function of the E protein of SARS-CoV-2. Furthermore, we observed that the P45L missense substitution in ORF7a which was found significantly associated with disease severity in our previous study [16], was not detected during the 4th wave of the pandemic when the Omicron variant was dominant in the country. This could be another reason why the mortality rate has decreased with the spread of the Omicron strain.

## Materials and methods

### Ethics committee approval

On February 1, 2021, this study developed as part of the implementation of the practical project No. A-IRV-2021-125 "Study of the genetics of SARS-CoV-2 coronavirus strains spread in the republic and creation of a distribution map in order to create the basis for the development of a COVID-19 vaccine". Approval for this study was granted by the Ethics Committee of the Center for Advanced Technologies of the Ministry of Higher Education, Science and Innovation on May 5, 2021 (approval number CAT-EC-2021/05-1). The patient information was gathered anonymously, and each patient was assigned a distinct ID number. Due to the anonymity of the collected data, a voluntary participation condition, and non-invasiveness of the sequencing experiment, along with a clear explanation of the purpose of sample collection to each participant, we received their verbal consent.

From April 1, 2022, this project was continued as project No. A-SS-202202113 entitled "Identification and study of the genotypes of SARS-CoV-2 coronavirus strains present in Uzbekistan and determination of the degree of their influence on the course of the disease" and was completed on March 31, 2023.

### Sample collection

Samples included in this study were obtained from COVID-19-diagnosed patients undergoing treatment at the State Hospital Zangiota No.1 and the Center for Sanitary and Epidemiological Service of the Republic of Uzbekistan. Nasopharyngeal and oropharyngeal specimens were collected in 3 mL of the viral transport medium (VTM) tube using flocked swabs and sent to the laboratory at a cold temperature (2–8˚C) within 72 hours post-collection. All the swab samples were stored at -80˚C until further analysis.

### RNA extraction

SARS-CoV-2 RNAs were extracted from patients' swabs using a Nucleic Acid Extraction and Purification Kit (Fosun Ultrapure NA) and QIAamp® Viral RNA Mini Handbook kit

(QIAGEN) according to the manufacturer's protocol. After viral extraction, RNAs were quantified through the Qubit ssRNA High Sensitivity Assay kit (Invitrogen by Thermo Fisher Scientific, Eugene, Oregon USA) on Qubit 2.0 Fluorometer (Life Technologies, Carlsbad, California, USA) to validate appropriate concentration.

## Real-time PCR for SARS-CoV-2

To detect SARS-CoV-2 viral infection, a one-step real-time PCR assay was performed using Biotest—SARS-CoV-2 RT-qPCR Kit (Biotest Lab LLC, Tashkent, Uzbekistan, patent FAP02010) developed by the team of Biotechnology Laboratory of Center for Advanced Technologies and Novel Coronavirus (2019-nCoV) RT-PCR detection kit (Shanghai, People's Republic of China) on the QuantStudio™ 5 real-time PCR System (Applied Biosystems, Foster City, USA) as per the manufacturer's instructions. The assay includes three viral targets ORF1ab (RdRp), N genes (N1, N2, N3) and E genes. Samples that tested positive for SARS-CoV-2 by RT-PCR with $C_T$ (cycle threshold) values $\leq 28$ were selected for this study.

## Quality control

To avoid contamination, RNA extraction and RT-qPCR were performed in separate rooms. A reagent negative control was also included during RNA extraction to account for any contamination during extraction. For each PCR and real-time PCR run, triplicate negative controls were included. All negative controls for RNA extraction and RT-qPCR were negative for the targets analyzed.

## Library preparation and sequencing

Whole genome amplification of the SARS-CoV-2 was performed using the following two kits: the CleanPlex® SARS-CoV-2 Research and Surveillance Panel (Paragon Genomics Inc., Hayward, CA, USA), and the Illumina COVIDSeq RUO Kit (Illumina, Inc., San Diego, CA, USA).

*CleanPlex*.SARS-CoV-2 Panel was used according to the manufacturer's instructions (version UG4001-04, Jan 2021 and UG4004-06 Feb 2022). Briefly, reverse transcription was performed using 5 µl (200 ng) previously extracted RNA followed by RT reaction purification by magnetic beads. Then, 5 µl purified RT reaction product was used for two multiplex PCR (in two non-overlapping SARS-CoV-2 target-specific primer pools, Paragon Genomics design) to amplify the whole SARS-CoV-2 genome. After post-mPCR (multiplex PCR) purification, digestion was performed to remove nonspecific PCR products, followed by post-digestion purification. In the next step, second PCR reaction was performed to amplify and add unique i5 and i7 dual indexes, on Veriti 96 Well Thermal Cycler (Applied Biosystem) with 24 cycles. Finally, the generated libraries were purified using (in all four purification steps) CleanMag® Magnetic Beads (Paragon Genomics Inc., Hayward, CA, USA) as per the manufacturer's instructions. Libraries were evaluated by gel-electrophoresis and considered for sequencing when a fragment size ~ 275 bp was obtained and were quantified using Qubit 2.0 dsDNA HS Assay Kit (Thermo Fisher Scientific, Waltham, MA, USA) to confirm that the concentration was above 2.0 ng/µl which is good quality for sequencing.

*Illumina COVIDSeq Test*. Illumina COVIDSeq RUO Kit was used according to the manufacturer's instructions (1000000126053 v05 Jun 2021). The first strand synthesis was carried out with 8.5 µL input RNA extracted from nasopharyngeal and oropharyngeal swabs, following the standard protocol. The synthesized cDNA was amplified in two separate PCR reactions (also including primers that target human RNA) within 35 cycles. The amplified PCR products were fragmented and tagged with adapter sequence using IDT for Illumina Nextera UD Indexes Set 1, 2, 3, 4 (384 indexes, 384 samples), followed by 7 cycles amplification of tagged

PCR amplicons. Then, pooling was performed by combining libraries from each 96-well plate into one tube. The generated pool was purified using Illumina Tune Beads as per protocols provided by the manufacturer (Illumina, Inc., San Diego, CA, USA). Finally, the purified pool was quantified using the Qubit 2.0 fluorometer dsDNA HS Assay Kit (Thermo Fisher Scientific, Waltham, MA, USA).

After confirmation of the library quality, the pooled libraries were further normalized to the final 4nM concentration and were denatured and neutralized with 0.2N NaOH and 400mM Tris-HCL (pH-8), following the Standard Normalization protocol on MiSeq System Denature and Dilute Libraries Guide (Illumina, San Diego, CA, USA) with a final denaturation and dilution to 12 pM. Paired-end sequencing 2×150 bp was performed on a MiSeq instrument (Illumina, San Diego, CA, USA) with Reagent Kit v3 (300 cycles), using 20 pM PhiX control spike-in of 5% for low-diversity libraries.

## Data and phylogenetic analysis

Standard bioinformatic tools were used to process the NGS raw data (FASTQ files) that were generated from MiSeq Local Run Manager (Illumina, San Diego, CA, USA). When *CleanPlex panel* was used, the NGS raw data (FASTQ files) were uploaded on the SOPHiA DDM platform (SOPHiA Genetics, Lausanne, Switzerland) for the external quality check, trimming of adaptors, variant call review, re-alignment of indels, quality measurements, and determination of the consensus genome by mapping to reference sequence MN908947 (NC 045512.2). For this, we have used a proprietary design pipeline to cover the entire genome.

DRAGEN COVIDSeq Test pipeline (Illumina Inc.) on the Illumina DRAGEN v3.6 Bio-IT platform was used for the row data generated from the COVIDSeq workflow as per standard protocol. The analysis involves sample sheet validation, data quality check, FASTQ generation, SARS-CoV-2 detection, and variant calling. For post-assembly quality control, a Phred score of $\geq$30 was applied, and sequences with a coverage depth of $\geq$30x were selected for further analysis. The Nextclade Web tool v2.14.1 was used to compare study sequences to SARS-CoV-2 reference sequences, assign them to clades, and determine their position within the SARS-CoV-2 phylogenetic tree [17].

Then the whole dataset of SARS-CoV-2 full genomes from Uzbek deposited in GISAID with accession numbers: EPI_ISL_3189000, EPI_ISL_3189001, EPI_ISL_3188999-EPI_ISL_3188963, EPI_ISL_18378658—EPI_ISL_18378665, EPI_ISL_15941879-EPI_ISL_15941918, EPI_ISL_15961449, EPI_ISL_15961450, EPI_ISL_18378666-EPI_ISL_18378686.

## Results

Clinical samples of nasopharyngeal and oropharyngeal swabs were collected from COVID-19 positive patients treated at Zangiota-1 and -2 State Hospital in Tashkent, Uzbekistan, and from the SES (Center for Sanitary and Epidemiological Services) of the Republic of Uzbekistan, during the four waves of the pandemic. SARS-CoV-2 positive samples (with Ct$\leq$28) were selected for further whole-genome sequencing. The samples covered in this article were sequenced using two amplicon-based approaches: 48 samples (out of 48 samples, 39 high-quality sequences were generated) using the Clean *Plex* SARS-CoV-2 Panel (Paragon Genomics) and 96 samples (out of 96 samples, 73 high-quality sequences were generated) using the Illumina *COVID Seq* Test (Illumina Inc). The acquired110 high-quality sequences, 47 of which were from 2021 (third wave) and 63 from 2022 (fourth wave), were analyzed in detail later in this article.

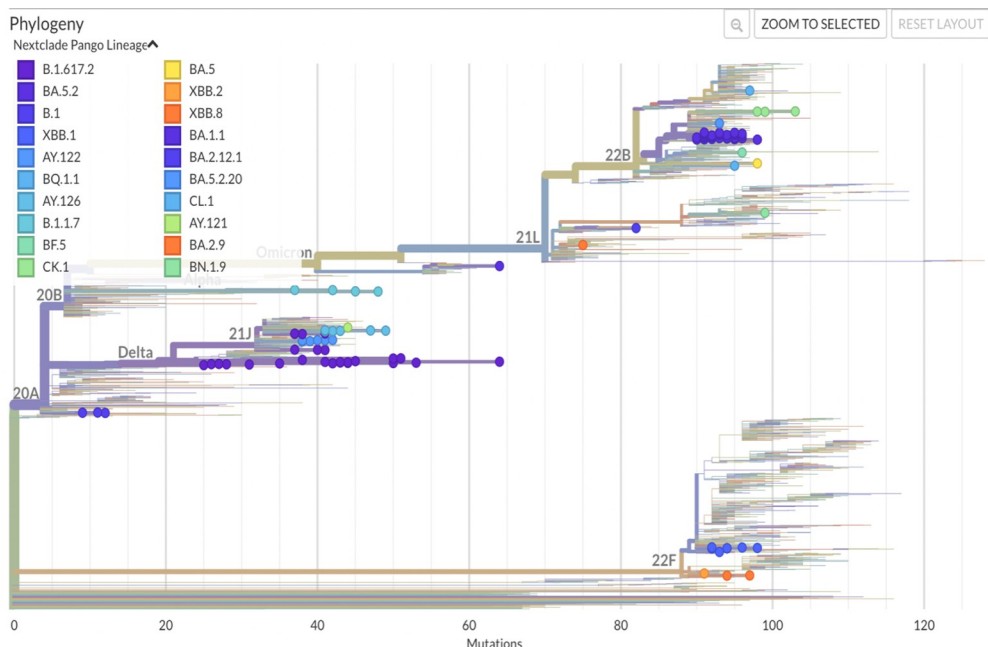

**Fig 3. Phylogenetic distribution of Uzbek SARS-CoV-2 genomes.** The phylogenetic tree was generated by Nextstrain. PANGOLIN lineages of SARS-CoV-2 are represented.

Based on the results of the sequence, a comparative analysis was carried out with the coronavirus variants common in the territory of Uzbekistan, Fig 3. From the previous reports it was determined that the 1st wave of the pandemic corresponds to the wild type of SARS-CoV-2- Wuhan strain [15], from early June 2020. Following the first wave, the Alpha (B.1.1.7) variant started to spread in the country as the 2nd outbreak, October 2020. Further analysis revealed that from the beginning of July 2021, the Delta (B.1.617.2) variant spread quickly and on a large scale as the 3rd wave in Uzbekistan, while the Omicron variant (B.1.1.529) spread as the 4th wave of COVID-19 pandemic, from the beginning of January 2022, Fig 1.

To get an insight into the genetic epidemiology, the genomes were analyzed for their phylogenetic distribution, Figs 3 and 4. When the genome Wuhan/WH01 (EPI_ISL_406798) was used as the reference for constructing the tree, the Nextclade-based phylogenetic analysis of the SARS-CoV-2 genome samples sequenced in our study revealed that the most currently distributed SARS-CoV-2 variants in dataset belong to the following clades: 20A, 20I, 21A, 21J, 21K, 21L, 22B, 22C, 22D, 22E and 22F clades, Fig 4.

By Nextstrain classification out of 110 studied SARS-CoV-2 samples, 61 genomes (55%) fell into 21K, 21L, 22B, 22C, 22D, 22E and 22F clades that represent the Omicron variant. In Fig 5, we summarized the phylogenetic distribution of the clades prorate to the lineages assigned by PANGOLIN.

Among these clades representing Omicron, the predominant occurrence (n = 37, 34%) is clade 22B (BA.5.2), which is a currently circulating sublineage of 21L Omicron with 22C, 22D and 22E, while the 8 genomes (7%) are grouped into clade 22F (XBB.2). The clades 21K (BA.1.1) and 21L (BA.2.9), Omicron sublineages emerged from the South Africa strain 21M (lineage B.1.1.529) [18], were also widespread in the country with the low-frequency rate 1% (GISAID IDs: EPI_ISL_15941881 and EPI_ISL_15941882).

Clades of Delta variant 21A and 21J (lineage B.1.617.2) were the next most common clades to spread in Uzbekistan with a frequency rate of 24% (n = 26), followed by clade Alpha 20I

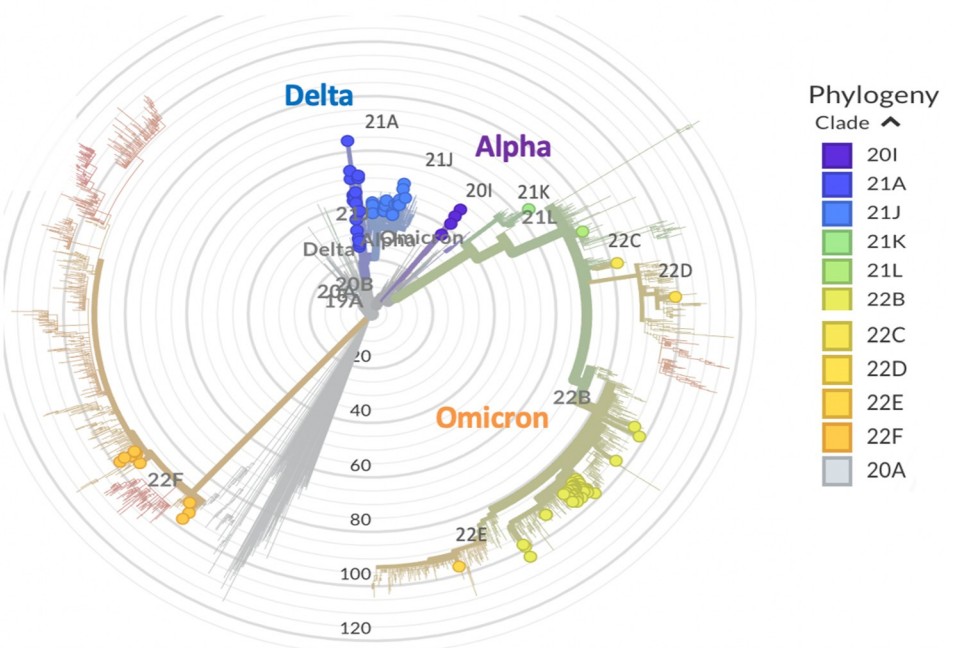

**Fig 4. Phylogenetic analysis of the SARS-CoV-2 coronavirus, common in the Republic of Uzbekistan clades by Nextstrain.** 110 genomes clustered under 20A, 20I, 21A, 21J, 21K, 21L, 22B, 22C, 22D, 22E and 22F Clades.

(lineage B.1.1.7). The Alpha 20I clade variants, namely EPI_ISL_3188965, EPI_ISL_3188992, EPI_ISL_3188994 and EPI_ISL_3188999 accounted for 4% of our SARS-CoV-2 sample genome sequences. We also identified the clade 20A (which emerged from 19A and was dominant during the European outbreak in March 2020 [18]) in our dataset, such as EPI_ISL_3188963, EPI_ISL_3188985 and EPI_ISL_3188990.

According to the results of genome-wide sequencing, the most common strains circulating in Uzbekistan belonged to the Delta (B.1.617.2) and Omicron (B.1.1.529) variants that made the third and fourth waves of the pandemic in 2021 and 2022, respectively, Fig 5. If from 2020 to November 2021 the lineages B.1 (20A), B.1.1.7 (Alpha), B.1.617.2 (Delta), AY.122 (Delta) were dominant and subdominant, from February 2022 the strain variants omicron BA.1.1, BA.2.12.1, BA.2.9, BQ.1.1, BN.1.9, BF.5, BA.5., BA.5.2, BA.5.2.20, XBB.1, XBB.2, XBB.8, CK., CL 1 became dominant in the country.

## Structural proteins (S-E-M-N)

Detailed analysis of amino acid substitutions in the SARS-CoV-2 genome showed that among the structural proteins (Spike, Envelop, Membrane and Nucleoprotein), the highest number of amino acid substitutions was found in the S gene, 115 changes. In the S protein, the SARS-CoV-2 mutation D614G was identified in all our isolates (n = 110; 100%), followed by T478K (n = 94; 86%), L452R (n = 79; 72%), P681H (n = 64; 58%), H655Y (n = 63; 57%), and G142D (n = 62; 56%). The increase in the number of mutations in the S gene can be explained by the emergence of the Omicron variant because as shown in Fig 6, when comparing the Omicron variant with the Delta variant, we can see that most of the S gene mutations are specific to the Omicron variant.

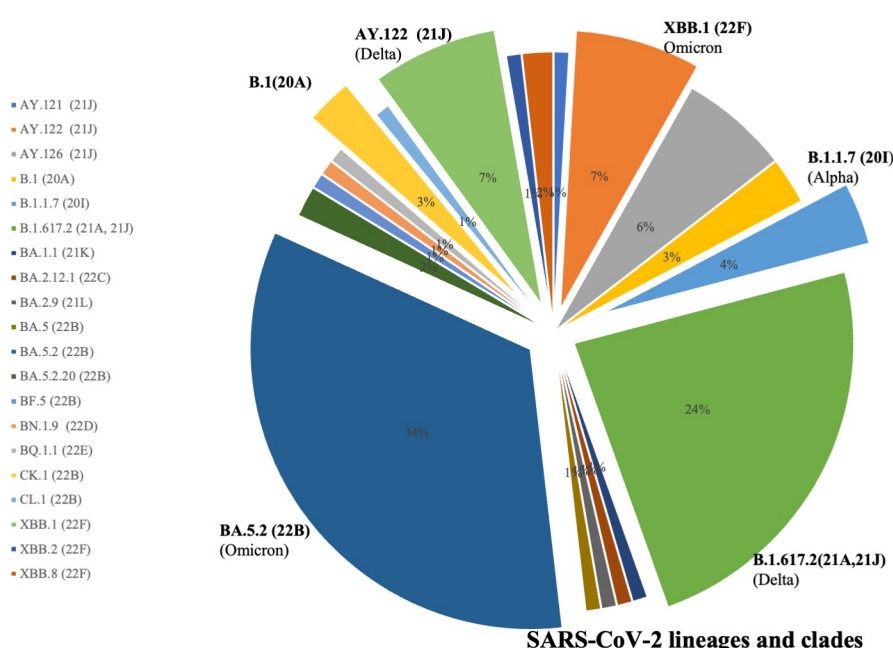

**Fig 5. SARS-CoV-2 common lineages and clades that circulating in Uzbekistan in 2021–2022.**

As we reported earlier, only one amino acid substitution—I82T was detected in the Membrane (M) protein, and we did not observe any mutation in the Envelope (E) protein [16]. In the present study, apart from the amino acid substitution I82T (n = 39; 35% of the isolates), we observed the following 4 changes: A63T and Q19E were detected in almost half of the isolates (n = 61; 55% and n = 60; 55%, respectively), D3N (n = 43; 39%) and D3G was detected in only one isolate. In contrast to their report, we identified the following 3 amino acid substitutions in Envelop (E) protein: T9I (n = 59; 54% of isolates), T11A (n = 11; 10%) and V58F amino acid

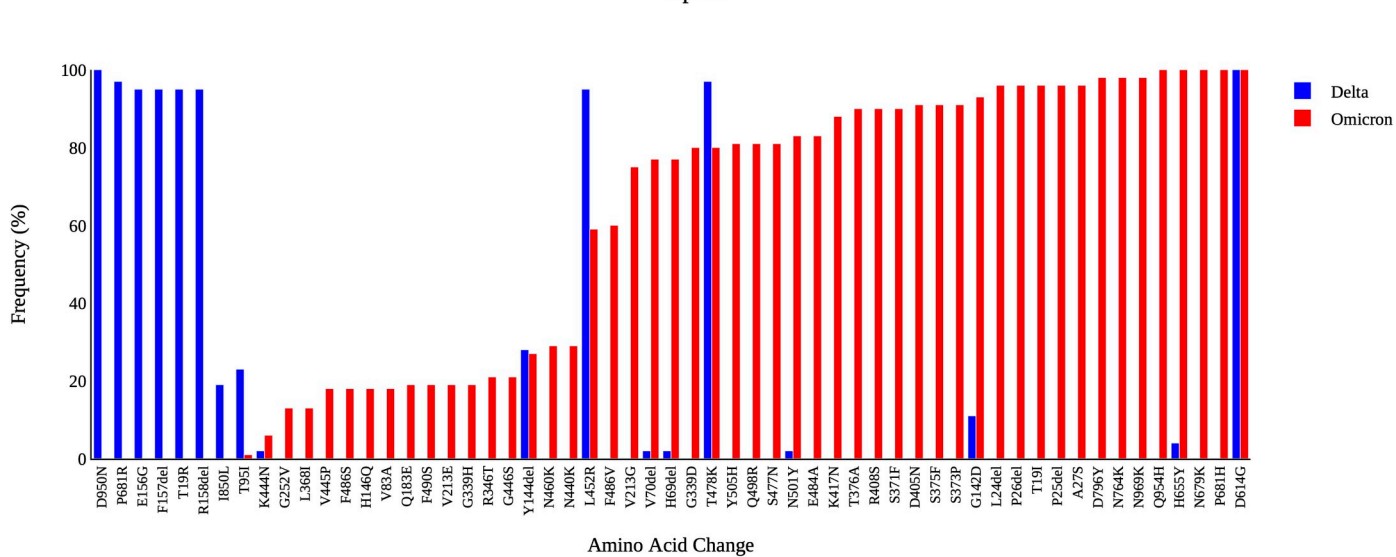

**Fig 6. High-frequency mutations detected in the spike gene of Delta and Omicron variants that were circulated in Uzbekistan.**

substitution in only one sample. This shows that as the pandemic progressed, the amino acid substitutions (in both structural and nonstructural proteins) have also increased. Compared to the E and M proteins, more mutations have occurred in the nucleoprotein, 25 amino acid substitutions. The most frequent of these, found in several isolates, were: G204R and R203K (n = 64; 58% of isolates) and P13L in 63 SARS-CoV-2 isolates, with a frequency of 57%. All amino acid substitutions in structural proteins (S-E-M-N) and the frequency of all identified missense mutations in SARS-CoV-2 isolates from our study are shown in Tables 1 and 2.

### Non—structural proteins (ORF1ab)

The other high-frequency mutations were identified in the ORF1ab region (113 and 46 mutations in ORF1a and ORF1b, respectively), which encodes 16 non-structural proteins (NSP1-NSP16). Of these 16 NSPs, the highest number of mutations was observed in NSP3, NSP2 and NSP13 with 41, 19 and 17 amino acid substitutions, respectively. The P323L substitution in NSP12 protein, which plays an important role in SARS-CoV-2 genome replication and transcription [19], was the most frequently detected substitution (identified in 98% of samples, n = 108), followed by the T492I substitution (n = 80; 73% of isolates) in NSP4. There were no mutations identified in the NSP11 gene. The all-amino acid substitutions of nonstructural proteins (ORF1ab) are illustrated in the S1 Table.

### Accessory proteins

During the four waves of the COVID-19 pandemic, mutations were observed in the genes for NS3, NS6, NS7a, NS7b and NS8. The most frequent amino acid substitutions were detected in NS3, and they were: T223I, S26L and K16T with high-frequency rates of 55% (n = 60), 38% (n = 42) and 6% (n = 7), respectively. A72T, G172R, Q185H, Q57R, E181K, F43C, G224C, H78Y, L41R, Q38R, T64I, T89I, V202L, Y107H, Y113H were identified in different samples with the low-frequency rate. Only three: I14V, T21I, and V24I substitutions in three different samples were detected in NS6. When NS7a had H47Y, L116F, P45L, R89I, T120I, V29L, V82A, NS7a had only I27T and T40I. NS8 had also several mutations, including stop codons at G8 (G8stop), K68 (K68stop) and Q27 (Q27stop) and F41C, I71F, R52I, T26I, T87I, Y73C, S1 Table.

## Discussion

The current study shows that in Uzbekistan four clear waves were identified during the entire period of the COVID-19 pandemic, Fig 1. Based on previous efforts to sequence SARS-CoV-2 samples from Uzbekistan, the first wave was thought to have been triggered by the spread of the original Wuhan variant in mid-2020 (June to September) [15]. The present research showed that the emerged Alpha variant resulted in the 2nd wave of pandemics, followed by the 3rd and 4th waves of Delta and Omicron variants, respectively. In each new wave, the infection rate steadily increased, Fig 1. The highest number of COVID-19 cases detected in Uzbekistan (n = 1478) was observed in January 2022 during the fourth outbreak by Omicron. On the contrary to increased transmission of SARS-CoV-2, the mortality rate has decreased with the spread of the Omicron strain, Fig 2.

The whole genome sequencing of samples selected during the four pandemic waves in Uzbekistan showed important mutations in the different parts of the genome. Some of the mutations affect the structure and properties of SARS-CoV-2 like transmissibility, virulence, and susceptibility to vaccines.

Amino acid substitution D614G became the most prevalent of all known SARS-CoV-2 mutations of the S protein (found in all 100% of our sequenced isolates), since its first

**Table 1. Amino acid changes detected in spike protein during the four waves of the SARS-CoV-2 pandemic in Uzbekistan.**

| Amino acid changes in Spike | Total number of sequenced samples—110 | Frequency | 20A | Alpha | Delta | Omicron |
|---|---|---|---|---|---|---|
| **D614G** | **110** | **100%** | **3** | **4** | **42** | **61** |
| T478K | 94 | 86% | 2 | 1 | 41 | 49 |
| L452R | 79 | 72% | 2 | 1 | 40 | 36 |
| P681H | 64 | 58% | 0 | 3 | 0 | 61 |
| H655Y | 63 | 57% | 0 | 0 | 2 | 61 |
| G142D | 62 | 56% | 0 | 0 | 5 | 57 |
| N679K, Q954H | 61 | 56% | 0 | 0 | 0 | 61 |
| D796Y, N764K, N969K | 60 | 55% | 0 | 0 | 0 | 60 |
| A27S, L24del, P25del, P26del, T19I | 59 | 54% | 0 | 0 | 0 | 59 |
| N501Y | 58 | 53% | 2 | 4 | 1 | 51 |
| D405N, S373P, S375F | 56 | 51% | 0 | 0 | 0 | 56 |
| R408S, S371F, T376A | 55 | 50% | 0 | 0 | 0 | 55 |
| K417N | 54 | 49% | 0 | 0 | 0 | 54 |
| H69del, V70del | 52 | 47% | 0 | 4 | 1 | 47 |
| E484A | 51 | 46% | 0 | 0 | 0 | 51 |
| Q498R, S477N, Y505H | 50 | 46% | 0 | 0 | 0 | 50 |
| G339D | 49 | 45% | 0 | 0 | 0 | 49 |
| V213G | 46 | 42% | 0 | 0 | 0 | 46 |
| E156G, F157del, R158del | 44 | 40% | 2 | 2 | 40 | 0 |
| D950N | 43 | 39% | 1 | 0 | 42 | 0 |
| P681R | 42 | 38% | 1 | 0 | 41 | 0 |
| T19R | 42 | 38% | 2 | 0 | 40 | 0 |
| F486V | 37 | 34% | 0 | 0 | 0 | 37 |
| Y144del | 36 | 33% | 3 | 4 | 12 | 17 |
| N440K, N460K | 18 | 16% | 0 | 0 | 0 | 18 |
| G446S, R346T | 13 | 12% | 0 | 0 | 0 | 13 |
| F490S, G339H, Q183E, V213E | 12 | 11% | 0 | 0 | 0 | 12 |
| F486S, H146Q, V445P, V83A | 11 | 10% | 0 | 0 | 0 | 11 |
| T95I | 11 | 10% | 0 | 0 | 10 | 1 |
| I850L | 8 | 7% | 0 | 0 | 8 | 0 |
| G252V, L368I | 8 | 7% | 0 | 0 | 0 | 8 |
| K444N | 5 | 5% | 0 | 0 | 1 | 4 |
| A570D, D1118H | 4 | 4% | 0 | 4 | 0 | 0 |
| T716I | 3 | 3% | 0 | 3 | 0 | 0 |
| S982A | 3 | 3% | 0 | 2 | 1 | 0 |
| A243del, A475V, E484V, L244del, N501T, S494P, T604N, T859N, | 2 | 2% | 0 | 0 | 2 | 0 |
| N450D, Q493R, V143del, Y145del | 2 | 2% | 0 | 0 | 0 | 2 |
| I1114T, S254F | 1 | 1% | 0 | 1 | 0 | 0 |
| D253A, E484Q, G1223S, I418V, L1063F, S680P, V1230L | 1 | 1% | 0 | 0 | 1 | 0 |
| A1020S, A67V, D253G, F157L, G496S, I210V, ins214EPE, K147E, K150D, K150R, K356T, K444T, L212I, L212S, L452Q, L455S, L752F, L981F, N211del, N856K, P631S, Q677H, R1014T, R21T, R237M, R346K, S151I, S255F, S371L, S704L, S940F, T547K, W152R | 1 | 1% | 0 | 0 | 0 | 1 |

occurrence in March 2020 [20]. As a missense mutation, D614G facilitates proteolysis at the furin cleavage site and alters the conformation of the RBD, leading to an increase in SARS-CoV-2 infectivity, transmissibility, and density of virion spikes [21–23]. In the Spike protein,

**Table 2. Amino acid changes detected in structural proteins (E-M-N) during the four waves of the SARS-CoV-2 pandemic in Uzbekistan.**

| | Amino acid changes in Structural Proteins (E-M-N) | Total number of samples 110 | Frequency | 20A | Alpha | Delta | Omicron |
|---|---|---|---|---|---|---|---|
| E | T9I | 59 | 54% | 0 | 0 | 0 | 59 |
| | T11A | 11 | 10% | 0 | 0 | 0 | 11 |
| | V58F | 1 | 1% | 0 | 0 | 0 | 1 |
| M | A63T | 61 | 55% | 0 | 0 | 0 | 61 |
| | Q19E | 60 | 55% | 0 | 0 | 0 | 60 |
| | D3N | 43 | 39% | 0 | 0 | 0 | 43 |
| | I82T | 39 | 35% | 0 | 0 | 39 | 0 |
| | D3G | 1 | 1% | 0 | 0 | 0 | 1 |
| N | G204R, R203K | 64 | 58% | 0 | 3 | 0 | 61 |
| | P13L | 63 | 57% | 0 | 0 | 2 | 61 |
| | E31del, R32del, S33del | 61 | 55% | 0 | 0 | 0 | 61 |
| | S413R | 56 | 51% | 0 | 0 | 0 | 56 |
| | D63G | 43 | 39% | 1 | 0 | 42 | 0 |
| | D377Y | 39 | 35% | 0 | 0 | 39 | 0 |
| | R203M | 38 | 35% | 0 | 1 | 37 | 0 |
| | G215C | 20 | 18% | 0 | 0 | 20 | 0 |
| | R385K | 10 | 9% | 0 | 0 | 10 | 0 |
| | T362I | 5 | 5% | 0 | 0 | 5 | 0 |
| | D3L, S235F | 4 | 4% | 0 | 4 | 0 | 0 |
| | P365S | 2 | 2% | 0 | 0 | 0 | 2 |
| | A156S, A182S, A182T, D144H, M234I, | 1 | 1% | 0 | 0 | 1 | 0 |
| | A218S, E136D, R40H, S37A | | | 0 | 0 | 0 | 1 |

the next high-frequency mutations were T478K and G142D (covering the 94–62% of samples sequenced) which can be found in both Omicron and Delta variants. Previous studies reported that the T478K mutation led to a higher binding affinity for human ACE2 with the following mutations: L452R, N501Y, S375F, S477N, S373P, Q498R, G339D, N440K, Q493R, and S371L that were identified in our samples [24, 25]. Two significant mutations: L452R and E484Q, also known as the spike double mutation, were first identified in samples from India. These unique mutations led to the emergence of a new Delta variant (B.1.617.2), which is classified as a VOC (Variants of Concern) by the WHO. Mutations specific to the Delta variant during the third pandemic wave in Uzbekistan L452R (72%), P681R (58%), and D950N (39%) were also identified in the sequence of Spike protein of studied samples [26].

The presence of the K417N mutation (49%) with the combination of R346K may reduce the neutralizing activity of the antibodies. Moreover, the R346K mutation which was only observed in one isolate hCoV-19/Uzbekistan/3_1465-CAT/2022|EPI_ISL_15941881, was reported as the only mutation that is distinctive to the sub-lineage BA.1.1 of Omicron [20, 27].

The most frequent mutations in E, M, and N proteins were T9I (54%), A63T (55%), and G204R, R203K (58%). The single point mutation T9I (T9I, threonine to isoleucine) in the envelope protein can influence the configuration of the E protein and ensure stronger anchoring to the viral membrane. Although the V58F mutation in the E-protein is not very common, it reduces the immune response of the B cells in the antigenic region [28–30]. Bingqing Xia et al, reported in their previous studies that T11A expression which was found in 11 of our isolates significantly alleviated cell death but did not alter the transfection efficiency and expression levels of SARS-CoV-2-E (envelop) protein [31].

The mutations Q19E and A63T in the membrane protein were not reported in our previous study [16] suggesting that they were only observed in all major Omicron subvariants (100%, in

61 Omicron variants) which were circulated during the 4th wave in Uzbekistan. And there is evidence about the impact of A63T which is the most frequent mutation on the stabilization of the M protein dimer in the studies by Anamica Hossain et al [32]. The N-terminal D3G mutation was present only in BA.1. 1 sub-lineage of Omicron variant (hCoV-19/Uzbekistan/3_1465-CAT/2022|EPI_ISL_15941881), and this mutation may affect the interactions with host cells [20, 32, 33].

Among the nucleocapsid mutations, the R203K and G204R mutations were the most common and these mutations made changes in the structure of the proteins [34]. Together, these mutations increase the viral load and the expression of sub-genomic RNA, additionally to SARS-CoV-2 replication, virulence, and pathogenicity [20, 22, 32, 35]. E31del, R32del, and S33del deletions in the N—terminal domain of nucleoprotein, may affect the assembly with M protein [36]. Some studies have indicated that the R203M mutation alone can enhance SARS-CoV-2 replication [37], while the P13L is associated with lower levels of transmissibility and death rates [38]. The impact of other nucleoprotein mutations on protein function has not been further studied well yet.

Non-structural proteins (NSP1-NSP16) encoded by the ORF1ab region of the SARS-CoV-2 genome are proteins that are not components of the virion but are transcribed and translated during host cell infection and play an important role in viral replication, translation, post-translational modification, assembly, evasion of the host immune system and other important functions [32]. Among these NSPs, NSP3, the largest protein of the virus, which plays an important role in virus replication, showed the highest number of mutations (n = 41), S1 Table. In NSP3, G489S, a mutation specific to Omicron, was observed with high frequency in all subvariants of Omicron detected in Uzbekistan, except in BA.1.1 (hCoV-19/Uzbekistan/3_1465-CAT/2022|EPI_ISL_15941881), as well as the S135R mutation in NSP1. However, in the Omicron BA.1.1 sub-variant, A1892T, I388T, K38R, L1266I mutations, and S1265 deletion were observed.

The mutation T492I in the NSP4 protein was observed in all sub-variants of Omicron (except BA.1), while the mutation L438F was only observed specifically in BA.2 and its descendant lineage BA.2.12.1. One study showed that these two mutations (L264F and T328I) might have functional significance in viral RNA replication [32]. NSP5 is the main protease that is important for SARS-CoV-2 as an antagonist of type I interferon (IFN) [12]. In this study, the mutation P132H in NSP5 which might affect enzyme activity, was observed in all subvariants of Omicron.

The next novel mutations which may play an important role in understanding of the COVID-19 disease, found in the NSP12 region of SARS-CoV-2 genome. NSP12 is necessary for the replication and transcription of the SARS-CoV-2 genome, as it encodes RNA-dependent RNA polymerase (RdRp) [16]. Previous studies reported that the P323L substitution of NSP12 that has been found in a large number of our samples (98%) could induce structural changes and adverse effect on proofreading during the replication of the SARS-CoV-2 genome [12]. The unique mutation R392C in NSP13 found in samples belonging to the Omicron variant and subvariants (except BA.1 subvariant) has been identified in this study.

NSP13 is a highly conserved RNA triphosphatase (RTPase), a vital coronavirus enzyme that unwinds double-stranded RNA in a 5′–3′ direction [32, 39, 40]. Anamica Hossain et al. suggested in their studies that the R392C which is located in the Rec1A domain may impact the NTPase activity (RNA Nucleoside Triphosphatase) of NSP13 helicase. Another study reported that this mutation may lead to a slight change in protein folding and alter the secondary structure of the protein [20].

The accessory proteins are important virulence factors involved in various pathogenesis mechanisms during SARS-CoV-2 infection. In addition, accessory proteins may play an

important role in immune evasion mechanisms, that enhance viral survival in the host system [32, 41]. The T223I mutation in NS3 is the largest accessory protein. Except in the BA.1 subvariant, this mutation (T223I) was present in all Omicron subvariants studied in this research. However, no significant effects have been noted for this mutation. In our previous study, we reported that the amino acid substitution ORF7a: P45L in the Delta isolates (n = 8, 7% of our current isolates) had a significant association with disease severity [16]. Our present study showed that P45L was not detected during the 4th wave of the pandemic when the Omicron variant was dominant in the country. This could be another reason why the mortality rate has decreased with the spread of the Omicron strain. Moreover, one study [42] suggested that the infection with the Omicron variant results in fewer long-COVID symptoms compared to previous variants (Alpha, Delta, and the historical variant-20A.EU2). As they indicated, after infection with SARS-CoV-2, post-COVID symptoms were observed in the individuals over the different periods, however, individuals infected with the historical variant (20A.EU2) were more likely to develop long-COVID symptomatology.

## Conclusion

Whole genome sequencing samples from patients with COVID-19 infection over the period of 2020–2022 using the above mentioned NGS methods allowed us to detect different variants (lineages and clades) of SARS-CoV-2 contributing to all four waves of COVID-19 pandemic. Our investigation showed that all viruses circulating during the 3rd wave and 4th waves belonged to the Delta and Omicron variants, respectively. Analysis of the mutations identified in our genome-wide study, showed that with the emergence of each new variant in our country, the COVID-19 pandemic has also progressed. This may be due to the considerable increase in the number of mutations (Alpha—46, Delta- 146, and Omicron—200 mutations were observed in studied samples) in each emerged variant that shows the SARS-CoV-2 evolution. Here, it is reasonable to mention that we have encountered several challenges during our research on SARS-CoV-2. When SARS-CoV-2 was first introduced to Uzbekistan, there was a shortage of test systems for widespread population testing. Consequently, during the initial pandemic wave, there was a global demand for reagents of genome sequencing, leading to high costs. As a consequence, our laboratory was unable to sequence a large number of samples compared to other research facilities. In addition, we struggled to maintain a consistent sample size across different waves. We believe that we would have achieved even better results if we had been able to sequence a larger number of samples and the same number of samples from each wave. Furthermore, during the pandemic, researchers have gained expertise in genome sequencing, and the laboratory has been well-equipped, leading to an increased level of preparedness for similar situations. We hoped that the detection of new variants that were circulated in Uzbekistan would have important implications for national health policy by enabling rapid tracking and investigation of infections in hospitals and communities and developing on-time alternative measures against them.

## Supporting information

**S1 Table. The substitutions of nonstructural proteins and accessory proteins during the four waves of SARS-CoV2 pandemic in Uzbekistan.**
(DOCX)

**S1 File. Genome sequences of 110 SARS-CoV-2 samples from Uzbekistan collected during four pandemic waves.**
(TXT)

**S2 File. SARS-CoV-2 genome annotation metadata of 110 samples from Uzbekistan collected during four pandemic waves.** Variants, lineages, clades, and all amino acid substitutions are listed.
(XLSX)

## Acknowledgments

The authors would like to express their great gratitude to the Republican Special Hospital No. 1 and No. 2 Zangiota and the Center for Sanitary and Epidemiological Services of the Republic of Uzbekistan for collecting the biological specimens and generating the metadata for GISAID. We also acknowledge all those who contributed to this research, i.e., the authors and the researchers.

## Author Contributions

**Data curation:** Gulnoza Esonova.

**Funding acquisition:** Shahlo Turdikulova, Dilbar Dalimova.

**Investigation:** Gulnoza Esonova, Abrorjon Abdurakhimov, Shakhnoza Ibragimova, Diyora Kurmaeva, Jakhongirbek Gulomov, Khonsuluv Sohibnazarova, Alisher Abdullaev.

**Methodology:** Gulnoza Esonova, Abrorjon Abdurakhimov, Alisher Abdullaev, Shahlo Turdikulova, Dilbar Dalimova.

**Resources:** Doniyor Mirazimov.

**Supervision:** Shahlo Turdikulova, Dilbar Dalimova.

**Visualization:** Gulnoza Esonova.

**Writing – original draft:** Gulnoza Esonova.

**Writing – review & editing:** Alisher Abdullaev.

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
