## [Decision Letter · Decision Letter 0]

20 Mar 2024

PONE-D-24-01894Complete genome sequencing of SARS-COV-2 strains that were circulating in Uzbekistan over the course of four pandemic wavesPLOS ONE

Dear Dr. Esonova,

Thank you for submitting your manuscript to PLOS ONE. After careful consideration, we feel that it has merit but does not fully meet PLOS ONE’s publication criteria as it currently stands. Therefore, we invite you to submit a revised version of the manuscript that addresses the points raised during the review process.

**Your manuscript, "Complete genome sequencing of SARS-COV-2 strains that were circulating in Uzbekistan over the course of four pandemic waves" (PONE-D-24-01894), has been assessed by our reviewers. Although it is of interest, we are unable to consider it for publication in its current form. The reviewers have raised a number of points that we believe would improve the manuscript and may allow a revised version to be published in PLOS ONE.**

We look forward to receiving your revised manuscript.

Kind regards,

Nihad A.M Al-Rashedi

Academic Editor

PLOS ONE

Journal Requirements:

This study has been supported by the research grant from the Ministry of Innovative Development, Republic of Uzbekistan (Research Grant number: А-IRV-2021-125). The funders had no role in study design, data collection and analysis, decision to publish, or preparation of the manuscript. No authors received award from this foundation.

3. Please amend the manuscript submission data (via Edit Submission) to include authors Dr. Abrorjon Abdurakhimov, Dr. Shakhnoza Ibragimova, Dr. Diyora Kurmayeva, Dr. Jakhongirbek Gulomov, Dr. Doniyor Mirazimov, Dr. Khonsuluv Sohibnazarova, Dr. Alisher Abdullaev, Dr. Shahlo Turdikulova and Dr. Dilbar Dalimova.

4. We are unable to open your Supporting Information file S1_File.fasta. Please kindly revise as necessary and re-upload.

Reviewers' comments:

Reviewer's Responses to Questions

**Comments to the Author**

1. Is the manuscript technically sound, and do the data support the conclusions?

Reviewer #1: Yes

Reviewer #2: Partly

2. Has the statistical analysis been performed appropriately and rigorously? 

Reviewer #1: Yes

Reviewer #2: Yes

3. Have the authors made all data underlying the findings in their manuscript fully available?

Reviewer #1: Yes

Reviewer #2: Yes

4. Is the manuscript presented in an intelligible fashion and written in standard English?

Reviewer #1: Yes

Reviewer #2: Yes

5. Review Comments to the Author

Reviewer #1: The study conducted whole genome sequence analysis of 110 SARS-CoV-2 genomes in Uzbekistan to elucidate the circulation of variants during the pandemic waves. Results revealed that the Omicron variant (B.1.1.529) predominated at 55%, followed by the Delta variant (B.1.617.2) at 38%. A total of 347 amino acid level variations were identified, with mutations primarily clustered in the ORF1ab and S gene regions, including the notable D614G mutation in the S gene. These findings indicate an adaptive evolution of SARS-CoV-2, with an increasing number of mutations over time potentially affecting transmissibility. While the study provides intriguing insights, I recommend further revisions to refine the manuscript:

1. In the introduction, it's essential to underscore that the emergence of new variants has facilitated the virus's evasion of detection using molecular assays (DOI: 10.1002/jmv.28241) and immune escape, leading to breakthrough infections post-vaccination (DOI: 10.1002/jmv.27688).

2. Kindly emphasize the novelty of the present study amidst the existing genomic characterization data for SARS-CoV-2 and its variants.

3. In the discussion section, highlight that efforts to detect new variants extended beyond clinical surveillance to include wastewater-based genome monitoring. This method rapidly detects viruses up to the strain level, enabling tracking of variant transmission routes and evolutionary dynamics (DOI: 10.1016/j.ijheh.2023.114224).

4. Discuss the necessity of characterizing emerging variants and their potential impact on long COVID development, now recognized as the most dreaded sequelae of acute SARS-CoV-2 infection. Reference studies such as DOI: 10.1016/j.jinf.2023.12.004 and DOI: 10.3390/v14122629.

5. Kindly address the limitations of the present study and provide recommendations to mitigate these limitations.

Reviewer #2: My specific comments are the followings:

Table 1: it is not known the type, total number, and the frequency of mutations detected in a specific VOC. The authors need to show the frequency of mutations within a specific VOC. As a reference, please see figure 3 of this publication: https://pubmed.ncbi.nlm.nih.gov/38244104/ Those should be the main table/ figure, and for the original table 1, you can move it as supplementary table because the information is not that informative (it combines all sequence data from the four waves).

Table 2: it is also similar to Table 1. The current presentation is not informative. It is not known which mutation that is highly frequent (>10%) and which mutation that is rarely found. You can move it as a supplementary table. It is not known which mutation that belonged to a specific VOV (for example, mutation that is exclusively found in Delta but not in Omicron variant).

Line 44: change “4” to “four”.

Line 96-98: The statement is not correct. The emergence of new strains is not associated with increased transmission and virulence of the new strains compared to previous strains.

Line 277-280: Is that because the majority of samples sequenced were from the fourth wave? So, it is due to sampling bias. The authors should provide the details how many samples out of 110 samples belonged to a specific wave.

Line 232-233: the correct one is “phylogenetic tree was generated by Nextstrain”.

Line 238, 241: the correct one is “wave”.

Line 352: the correct one is “affect”.

Line 356: D614G is not a silent mutation, please revise.

Line 360: : the correct one is “found”.

Line 378-380: in what protein?

Line 425: change to “variant and subvariants”.

Line 56-59 and 451-452: which data supports this statement?

Be consistent with “Covid-19” in the whole manuscript since different writings were identified, such as COVID-19 or covid-19.

Table 1, for the frequency, change into percentage (for example: 1,000 to 100%)

In addition, the article still needs an English editing by professionals.

6. PLOS authors have the option to publish the peer review history of their article (what does this mean?). If published, this will include your full peer review and any attached files.

Reviewer #1: No

Reviewer #2: **Yes: **Mohamad S. Hakim, PhD.

---

## [Author Response · Author response to Decision Letter 0]

30 Apr 2024

Respond to the Journal Requirements: 

1. When we wrote and revised the manuscript, we tried to follow the requirements of the PLOS ONE journal.

2. This study has been supported by a research grant from the Ministry of Innovative Development, Republic of Uzbekistan (Research Grant number: А-IRV-2021-125). The funders had no role in study design, data collection and analysis, decision to publish, or preparation of the manuscript. No authors received an award from this foundation.

We included the Funding Statement within our cover letter and reuploaded it.

Please let us know if you need any further information.

3. We changed the manuscript submission data to include authors.

4. We have changed the format of the Supporting Information file S1_File.fasta, allowing it to open easily.

Response to Reviewers:

Response to Reviewer #1

Dear reviewer, 

We appreciate your thoughtful feedback on our article. Your comments were insightful, and we tried to cover them as much as we can. Below are our responses to the points you raised.

1. In the introduction, it's essential to underscore that the emergence of new variants has facilitated the virus's evasion of detection using molecular assays (DOI: 10.1002/jmv.28241) and immune escape, leading to breakthrough infections post-vaccination (DOI: 10.1002/jmv.27688).

Our response:

We appreciate your suggestion regarding the introduction section. After reviewing the articles you recommended, we have tried to incorporate additional information on lines 116-118 in the Manuscript file (139-141 in the Revised Manuscript with Track Changes file), taking into account the focus of our research.

2. Kindly emphasize the novelty of the present study amidst the existing genomic characterization data for SARS-CoV-2 and its variants.

Our response:

Based on our obtained results, we tried to highlight the novelty of our present study on lines 52-54 in the Manuscript file (54-56 in the Revised Manuscript with Track Changes file).

3. In the discussion section, highlight that efforts to detect new variants extended beyond clinical surveillance to include wastewater-based genome monitoring. This method rapidly detects viruses up to the strain level, enabling tracking of variant transmission routes and evolutionary dynamics (DOI: 10.1016/j.ijheh.2023.114224).

Our response:

We appreciate the reviewer for emphasizing the significance of wastewater-based genomic surveillance. After reviewing the suggested article, we considered that wastewater-based genome monitoring would be one possible method that helps to increase the pandemic monitoring in Uzbekistan and preferred to highlight it in our manuscript on lines 463-467 in the Manuscript file (612-616 in the Revised Manuscript with Track Changes file).

4. Discuss the necessity of characterizing emerging variants and their potential impact on long COVID development, now recognized as the most dreaded sequelae of acute SARS-CoV-2 infection. Reference studies such as DOI: 10.1016/j.jinf.2023.12.004 and DOI: 10.3390/v14122629.

Our response:

After reviewing the articles you suggested, we have tried to give a brief information about long-COVID symptoms on 457-462 lines in the Manuscript file (606-611 in the Revised Manuscript with Track Changes file).

5. Kindly address the limitations of the present study and provide recommendations to mitigate these limitations.

Our response:

Thank you so much for your feedback, as we found it very reasonable. In the conclusion section (478-588 lines in the Manuscript file, 627-637 in the Revised Manuscript with Track Changes file), we have made an effort to address the limitations of our research and provide our insights on them.

Please, check the revision and suggest your opinion.

Response to Reviewer #2

Dear Mohamad S. Hakim,

We want to express our gratitude for the insightful feedback you provided on our article. Your comments were not only reasonable but also so valuable that we have made an effort to integrate them into our work. Please find below our responses to the comments you raised.

Table 1: it is not known the type, total number, and the frequency of mutations detected in a specific VOC. The authors need to show the frequency of mutations within a specific VOC. As a reference, please see figure 3 of this publication: https://pubmed.ncbi.nlm.nih.gov/38244104/ Those should be the main table/ figure, and for the original table 1, you can move it as supplementary table because the information is not that informative (it combines all sequence data from the four waves).

Our response:

Thank you very much for your feedback on Table 1. After reviewing the suggested article, we found it helpful in creating a clear visualization to display the frequency of mutations found in our sequenced Delta and Omicron variants. Following your advice, we attempted to detail the mutations identified in a particular VOC in Table 1. As a result, Table 1 was reorganized, however, we would like to keep it in the manuscript 302-308 and 325-327 in (373-379 and 413-420 in the Revised Manuscript with Track Changes file).

Please, check this version and suggest your opinion.

Table 2: it is also similar to Table 1. The current presentation is not informative. It is not known which mutation that is highly frequent (>10%) and which mutation that is rarely found. You can move it as a supplementary table. It is not known which mutation that belonged to a specific VOV (for example, mutation that is exclusively found in Delta but not in Omicron variant).

Our response:

As you suggested we have rearranged Table 2 and move it to supplementary material.

Line 96-98: The statement is not correct. The emergence of new strains is not associated with increased transmission and virulence of the new strains compared to previous strains.

Our response:

After carefully read the statement, we realized that it is not applicable to all variants of SARS-CoV-2 and therefore inaccurate. Thus, we prefer to remove it.

Line 277-280: Is that because the majority of samples sequenced were from the fourth wave? So, it is due to sampling bias. The authors should provide the details how many samples out of 110 samples belonged to a specific wave. 

 Our response:

Thank you so much for your comment as we found it very reasonable. We tried to get the same number of samples from 2021 and 2022, however, out of 144 samples, we obtained 110 high-quality sequencing results (20A-3, Alpha-4, Delta-42, and Omicron-61) for analysis. As we could not take, the same number of samples from each wave, we acknowledge that we cannot assert that "the most common strains circulating in Uzbekistan belong to the Omicron variant...". Therefore, we have included information about the distribution of samples from each period (wave) on lines 242-244 in the Manuscript file (299-301 in the Revised Manuscript with Track Changes file).

We have reviewed and addressed the comments on the lines you have mentioned: Line 44, Line 232-233, Line 238, 241, Line 352, Line 356, Line 360, Line 378-380, Line 425, Line 56-59, 451-452, and the final ones. We have made efforts to correct them and incorporate the necessary revisions.

Once again, we express our sincere gratitude to the reviewers.

Kind regards,

Esonova Gulnoza

 PhD student and junior researcher

 Biotechnology Laboratory of Center for Advanced Technologies.

esonovagulnoza96@gmail.com

---

## [Decision Letter · Decision Letter 1]

1 Jul 2024

PONE-D-24-01894R1Complete genome sequencing of SARS-COV-2 strains that were circulating in Uzbekistan over the course of four pandemic wavesPLOS ONE

Dear Dr. Esonova,

Thank you for submitting your manuscript to PLOS ONE. After careful consideration, we feel that it has merit but does not fully meet PLOS ONE’s publication criteria as it currently stands. Therefore, we invite you to submit a revised version of the manuscript that addresses the points raised during the review process.

We look forward to receiving your revised manuscript.

Kind regards,

Nihad A.M Al-Rashedi

Academic Editor

PLOS ONE

Journal Requirements:

Reviewers' comments:

Reviewer's Responses to Questions

**Comments to the Author**

1. If the authors have adequately addressed your comments raised in a previous round of review and you feel that this manuscript is now acceptable for publication, you may indicate that here to bypass the “Comments to the Author” section, enter your conflict of interest statement in the “Confidential to Editor” section, and submit your "Accept" recommendation.

Reviewer #3: (No Response)

Reviewer #4: (No Response)

2. Is the manuscript technically sound, and do the data support the conclusions?

Reviewer #3: Partly

Reviewer #4: Yes

3. Has the statistical analysis been performed appropriately and rigorously? 

Reviewer #3: N/A

Reviewer #4: N/A

4. Have the authors made all data underlying the findings in their manuscript fully available?

Reviewer #3: Yes

Reviewer #4: Yes

5. Is the manuscript presented in an intelligible fashion and written in standard English?

Reviewer #3: No

Reviewer #4: Yes

6. Review Comments to the Author

Reviewer #3: Major concern:

L221, “Standard bioinformatic tools” is not quite specific enough. Obviously DRAGEN can analyze the samples prepared with the COVIDSeq workflow, however, the DRAGEN pipeline would not be appropriate to analyze the results from the CleanPlex (Paragon) workflow. Specifically, the Paragon primer sequences should be removed/masked from the reads prior to read mapping or assembly to prevent false base calls. Regardless of bioinformatic pipeline, it would be nice to know a little more about the pipeline – quality trimming, host removal, read mapping, consensus calling – even if DRAGEN was used. The authors state elsewhere that only “high quality” sequences were used (e.g., L240, L241), and so they should specifically state the post-assembly quality control implemented. E.g., was it % coverage? At what depth? With/out ambiguous base calls? Was it determined in the bioinformatic pipeline or in the analysis pipeline (e.g., nextclade quality score?) This is quite important, as some of the sequences that were deposited in GISAID show a relatively high level of unlabeled private mutations and reversions, resulting in some long branch lengths relative to reference strains. This could obviously be true, but it does not exclude the possibility that these are simply the result of inappropriate bioinformatics (e.g., unmasked primer sequences, index hopping, or contamination). The authors have provided info about which samples were processed with the COVIDSeq workflow in GISAID – and I assume consensus sequences were generated with DRAGEN – but no information was provided about the CleanPlex workflow. Some of the sequences were stated in GISAID to be assembled with UGENE v.39.0 – were these reads generated with the CleanPlex workflow? And, if so, this should be described in the manuscript. The main issues with sequencing data have been known/understood since very early in the pandemic (e.g., https://virological.org/t/issues-with-sars-cov-2-sequencing-data/473), and have been demonstrated with WGS ring tests/EQAs/proficiency testing (there are a few publications now). However, the quality of the sequencing is not mentioned in this manuscript as a limitation of the analysis, and might be mentioned in this manuscript.

Similarly, the second major concern is the discussion of limitations.

The authors state their aim (L120) was “to provide insights into the spread of infection, evolutionary patterns, and genetic diversity of the virus to enable effective management and preventive measures in Uzbekistan.” They report on genetic diversity and specific mutations detected in a subset of circulating strains sequenced from in-patients (?). Their approach is whole genome sequencing – which is appropriate, however, there are (practical, sampling, and methodological) limitations with this method to capture genetic diversity and detect specific mutations (e.g., see ref 1, ref 2, ref 3, etc). The authors should probably discuss these as they relate to the main findings of their study (i.e., frequency of mutations in specific lineages), and how they relate to effective management/preventive measures in [a country].

The “Conclusion” section could be re-labeled “Limitations,” but the authors should discuss how these practical challenges specifically affected the data presented in this manuscript. In this section, they discuss the challenges to onboarding NGS in routine virus diagnostics, so they could mention how this affected their confidence in calculating the frequencies of specific mutations and/or lineages (e.g., how/why did they only select "high quality sequences" for comparison). I think the “retrospective” narrative of best practices and experiences is really nice, and very appropriate for this manuscript, given that the COVID-19 pandemic expanding capacity for virus sequencing around the globe. So please keep this, but it doesn’t really address the limitations of the study as presented.

L451, here might be a good time to mention the limitation/strength of this study in relation to previous studies. E.g., identifying ORF7a:P45L as a virulence determinant – was the sample size better in this study, or was it simply because another variant was prevalent? What was the evidence that virulence was decreased with Omicron (presented here or cited elsewhere)?

Another example of a missed opportunity to discuss the limitations with respect to the findings in this study is the new paragraph on line 463. This paragraph was requested by Reviewer 1, but the point is not presented with respect to the findings of this study. There are problems with wastewater-based surveillance in accurately identifying “strain” composition (major variants, at best), and – as far as I know – this technique has never been capable of tracking transmission routes with molecular epidemiology (unless I am misunderstanding what the authors are intending to convey – fecal-oral route?). The study cited [ref 44] does not address these limitations in their analysis, and I would not cite it as an example of how wastewater is useful in surveillance of viruses. Consider the complicated analysis presented by Amman et al. (2002, Nat Biotech) to deconvolute sequencing data and semi-accurately assign lineages, and consider that ref 44 did not take any of that into account. I think the point is valid, that if the authors wished to survey the genetic diversity of circulating strains, an alternative to testing symptomatic patients is environmental sampling; however, patient sampling provides more accurate genetic detail at the individual level.

There are a few minor concerns, which should be considered strong suggestions for clarification to reduce ambiguity or correct inaccuracies:

1. L106, why is May 2021 not considered a “peak”?

2. L124, this is a stylistic suggestion, but I believe the introduction could end here (with “…preventive measures in Uzbekistan.”) The rest of the paragraph lists specific findings that are included in the results; and, in particular, the last sentence is discussion/conclusion without direct evidence presented in this study.

3. L132, P45L is ambiguous – which gene/orf? (ORF7a?) Check to make sure that all substitutions/indels are labeled as "gene:A##B", or that their reference gene is otherwise unambiguous.

4. L176-177, I hope the authors realize that the “RT” in “RT-PCR” stands for “reverse transcription” and the “q” in “RT-qPCR” indicates it is a “real-time” assay, even if the result is semi-quantitative. Therefore, in this case, “RT-qPCR” is “real-time reverse transcription PCR”. See the MIQE guidelines. It seems clear in L188-189, but here it is a bit conflated.

5. L183 & L186, what is the difference between “CleanMag®” and “CleanPlex”?

6. L302 (“The increase in the number…”), L317 (“This shows that…”) and elsewhere: these are discussion points.

7. Table 1 is now two tables, somehow - the authors should fix the formatting. Note that the some rows of the “Spike” portion of Table 1 contain duplicated sequence features.

8. L332 and elsewhere – is this nsp12: P314L? nsp12:K323 has been largely unchanged. I think the authors should re-evaluate their conclusions about this mutation. The website they cite (ref. 14, not a peer-reviewed scientific source) does not mention this mutation. However, there have been several peer-reviewed scientific publications that discuss this specific mutation.

9. L358, where is the increased “rate” of infection shown?

10. L437, I am confused by this sentence and the following paragraph. nsp13:R392C was “reported for the first time in this study”, but was previously studied [ref 41]? Actually, ref 41 (Kuman et al., published since March 2023 in Vacunas, please update the reference) does discuss nsp13, but does not identify this mutation. However, the authors seem to be referring to Hossain et al. 2022 (Microb Pathog) [ref 34], which discusses this mutation in nsp13 in their review of “all” nonstructural mutations associated with Omicron (although they are not clear about how they identified the mutations),

Please double-check all references, that (1) they are correctly cited in the text, (2) they are up-to-date, and (3) they are appropriate. Specifically:

Is [ref 8] necessary? They have already cited the source [ref 7]

Ref [13] is not an appropriate citation of the GISAID clade nomenclature

Ref [14] is not a peer-reviewed scientific manuscript

It is not required, but the manuscript would benefit from a lot of editing. Below, I list just a few suggestions from the introduction, but the entire manuscript would be much better with English language editing.

L41 “…conducted whole genome sequencing (WGS) analysis…”

L42, already defined WGS

“respectively” (correct on L357), and not “appropriately” (L47), “correspondingly” (L332), nor “properly” (L342)

L48, “variants” not “variations”

L49, “…followed by the S gene...”

L53, “…envelope (E) protein. In contrast, in our present study, we…” (same mistake is repeated in line 129)

L54, did you test the “structure and function”?

L56 & L475, “progressed” is ambiguous.

only capitalize proper nouns: L60 severe acute respiratory syndrome coronavirus 2; L86 phylogenetic assignment of named global outbreak lineages (move ref 11 here); L167 and L179 “real-time”

L70, “metagenomic” is probably more appropriate then “metatranscriptomics”

L74, “As sequencing is essential…”

L78, “…had quickly begun…”

L81, “newest” is obviously relative if this paper is published in 2024; and the recombinant genotypes are being ignored here.

L173, “Samples that tested positive for SARS-CoV-2…"

L174, probably should clarify that Ct = cycle threshold

L176-177, maybe re-word this sentence for clarity: “To avoid contamination, RNA extraction and RT-PCR amplification were performed in separate rooms.”

L272, “predominant”

L376, I suggest “samples from India”

L391, “…in 11 of our isolates…” (also, were these isolates?)

L392, seems to be missing a word (e.g., “…significantly alleviated cell death but did not alter…”)?

L394, beginning here, why point out amino acid abbreviations here and nowhere previously? I think it is unnecessary.

L397, is a sentence fragment with strange comma splicing

L423, NSP or Nsp? L471 COVID or Covid?

L449, reword “…that can be met…”

Reviewer #4: 1. The authors must be commended for addressing the comments raised by the previous reviewers.

2. This is an important article showing the genomic mutations of SARS-COV-2 strains circulating in Uzbekistan. However, the title does not (really) convey the content and findings. Should the title not read, "Complete genome sequencing of SARS-COV-2 strains that were circulating in Uzbekistan during the third and fourth pandemic waves in 2021 and 2022"? The authors state, "We tried to get the same number of samples from 2021 and 2022", and later refer to waves 3 and 4. This supports my recommendation to change the title.

3. The previous reviewers commented: "Line 96-98: The statement is not correct. The emergence of new strains is not associated with increased transmission and virulence of the new strains compared to previous strains." The authors responded, "After carefully read the statement, we realized that it is not applicable to all variants of

SARS-CoV-2 and therefore inaccurate [sic]. Thus, we prefer to remove it." However, I believe that this is an important concept and I advise the authors not to remove the statement, but rather to amend it, eg "The emergence of new strains is more often associated with increased transmission, but not necessarily incfreased virulence of the new strains compared to previous strains" see:https://pubmed.ncbi.nlm.nih.gov/36591708/ and https://www.nature.com/articles/s41579-023-00878-2

7. PLOS authors have the option to publish the peer review history of their article (what does this mean?). If published, this will include your full peer review and any attached files.

Reviewer #3: No

Reviewer #4: **Yes: **Burtram Clinton Fielding

---

## [Author Response · Author response to Decision Letter 1]

11 Aug 2024

We have addressed the reviewers' comments in our "Response to Reviewers" file. Regarding the editor's comment about the reference list, we carefully reviewed and rearranged it. We have noted the changes in the rebuttal letter.

---

## [Editor Report · Decision Letter 2]

15 Aug 2024

Complete genome sequencing of SARS-CoV-2 strains that were circulating in Uzbekistan over the course of four pandemic waves

PONE-D-24-01894R2

Dear Dr. Esonova,

We’re pleased to inform you that your manuscript has been judged scientifically suitable for publication and will be formally accepted for publication once it meets all outstanding technical requirements.

Kind regards,

Nihad A.M Al-Rashedi

Academic Editor

PLOS ONE
---

## [Editor Report · Acceptance letter]

22 Aug 2024

PONE-D-24-01894R2 

PLOS ONE

Dear Dr. Esonova, 

I'm pleased to inform you that your manuscript has been deemed suitable for publication in PLOS ONE. Congratulations! Your manuscript is now being handed over to our production team.

Kind regards, 

on behalf of

Dr. Nihad A.M Al-Rashedi 

Academic Editor

PLOS ONE